# Hereditary Diffuse Gastric Cancer: Molecular Genetics, Biological Mechanisms and Current Therapeutic Approaches

**DOI:** 10.3390/ijms23147821

**Published:** 2022-07-15

**Authors:** Lidia-Sabina Cosma, Sophie Schlosser, Hauke C. Tews, Martina Müller, Arne Kandulski

**Affiliations:** Department and Outpatients Department of Internal Medicine I, University Hospital Regensburg, 93053 Regensburg, Germany; lidia-sabina.cosma@ukr.de (L.-S.C.); sophie.schlosser@ukr.de (S.S.); hauke.tews@ukr.de (H.C.T.); martina.mueller-schilling@ukr.de (M.M.)

**Keywords:** hereditary diffuse gastric cancer (HDGC), *CDH1* germline mutation, *CTNNA1* mutation, molecular genetics

## Abstract

Hereditary diffuse gastric cancer is an autosomal dominant syndrome characterized by a high prevalence of diffuse gastric cancer and lobular breast cancer. It is caused by inactivating mutations in the tumor suppressor gene *CDH1*. Genetic testing technologies have become more efficient over the years, also enabling the discovery of other susceptibility genes for gastric cancer, such as *CTNNA1* among the most important genes. The diagnosis of pathogenic variant carriers with an increased risk of developing gastric cancer is a selection process involving a multidisciplinary team. To achieve optimal long-term results, it requires shared decision-making in risk management. In this review, we present a synopsis of the molecular changes and current therapeutic approaches in HDGC based on the current literature.

## 1. Introduction

Hereditary diffuse gastric cancer (HDGC) is a cancer syndrome characterized by a high prevalence of diffuse gastric cancer (DGC) and lobular breast cancer (LBC). It was first described in a New Zealand Māori family in 1998 [1].

Gastric cancer ranks as the fifth cause of cancer mortality worldwide after lung, colorectal, breast, and prostate cancer [2]. Approximately 1–3% of gastric cancers are hereditary [1,3]. Worldwide, a population incidence of approximately 5–10/100.000 births is estimated for HDGC [4]. Both the overall incidence and mortality of non-cardia gastric cancer have declined over the past four decades. However, the incidence is increasing among persons younger than 50 years [5]. Among the young, gastric cancer is associated with a high incidence of poor differentiation and signet ring cell morphology. In this patient cohort, gastric cancer presents in advanced stages at first diagnosis with poor survival rates even with surgical resection [6].

Approximately 40% of HDGC families exhibit germline mutations within the *CDH1* gene (cadherin 1) [4]. *CDH1* encodes for E-cadherin, a homophilic transmembrane protein with a tumor suppressor function that is localized to the adherens junctions in epithelial tissue. In normal cells, E-cadherin expression is crucial for cell–cell adhesion, cell mechanosensitivity, epithelial-mesenchymal transition and signaling for contact inhibition of cell proliferation [7].

Blair et al. published an update of the International Gastric Cancer Linkage Consortium (IGCLC) clinical management guidelines [8]. According to these 2020 IGCLC guidelines, HDGC is defined either by the presence of a *CDH1* or *CTNNA1* (catenin alpha 1) pathogenic variant in an isolated case of diffuse gastric cancer (DGC) or a family with two or more cases of DGC in first- or second-degree relatives [8]. With regard to HDGC screening, it remains clinically challenging to identify patients with a high pretest probability of HDGC and to screen for a *CDH1* or *CTNNA1* mutation. A *CDH1* test is recommended if defined criteria of the IGCLC are met. Individuals who fulfil criteria for genetic testing but are found to be negative for a *CDH1* variant should be subsequently tested for *CTNNA1*.

The clinical phenotype of HDGC shows considerable heterogeneity regarding the histomorphological type of cancer and age of onset [4,9]. The histopathology of advanced HDGC is comparable to sporadic DGC, although the presence of typical precursor lesions, in situ or pagetoid signet-ring cells, are very specific for *CDH1*-mutation related HDGC [10]. Early HDGC has an indolent phenotype, while advanced HDGC displays an aggressive phenotype with a mixture of pleomorphic cells, increased proliferation, and aberrant p53 expression [10,11].

If *CDH1* mutation is detected without the phenotypical manifestation of gastric cancer, it is an ongoing debate whether to recommend surgical resection or endoscopic control. *CDH1* mutation carriers harbor an approximately 70% lifetime risk to develop gastric cancer for male gender and 56% in women [4]. Therefore, *CDH1* variant carriers from confirmed HDGC families should be advised to consider prophylactic total gastrectomy (PTG), regardless of endoscopic findings [8]. Surgery is recommended in early adulthood, generally between 20 and 30 years of age [10]. PTG is not recommended for patients over 70 years unless there are significant mitigating circumstances. For those declining or wishing to postpone PTG, annual endoscopy carried out by experienced endoscopists with knowledge of HDGC is recommended [8,10]. The decision to perform PTG or continue with surveillance (endoscopic screening and biopsy) is influenced by a number of different and interrelated factors from the patients like objective risk confirmation, perceived familial cancer burden, subjective risk perceptions, experiences, and perceptions of the different risk management options and life stage [12].

## 2. Molecular Genetics and Histopathological Alterations in HDGC

### 2.1. Driving Mutations and Cancer-Predisposing Genes for HDGC

As introduced, several driver mutations and mutations of cancer-predisposing genes are known in HDGC. *CDH1* mutations are the most common germline mutations detected in gastric cancer and underlie HDGC syndrome. On the global level, approximately 30–40% of cases that fit the clinical criteria for HDGC carry a pathogenic variant in the germline *CDH1* gene [3,4,10]. *CDH1* encodes for the cell-to-cell adhesion protein E-cadherin [13]. Moreover, several *CDH1* germline mutations have been identified in different ethnic groups [9,14].

In patients with a strong HDGC family history but without *CDH1* mutations, other genetic causes have been described. The rarity of patients with HDGC without pathogenic *CDH1* variants makes the collection of large datasets challenging. Germline mutations of some related genes, such as *CTNNA1*, *MAP3K6*, *INSR*, *FBXO24*, *DOT1L*, *CD44*, *PALB2*, *MSH2*, *BRCA1*, *RAD51C*, and *MET* are susceptible to specific HDGC families (see Table 1) [15,16,17].

*CTNNA1* encodes for Catenin alpha-1, a *CDH1*-binding partner. The first HDGC family carrying a *CTNNA1* germline mutation was reported in 2013 [15]. Since then, four additional families have been reported, establishing *CTNNA1* as a second HDGC susceptibility gene in addition to *CDH1* [18]. However, little is known about the penetrance of pathogenic *CTNNA1* variants [19]. Nevertheless, the 2020 IGCLC clinical criteria apply and advocate the indication for genetic testing for *CTNNA1* in addition to genetic testing for *CDH1* [8].

Mutations in homologous recombination (HR) repair genes, such as *PALB2*, likely explain a significant fraction of inherited gastric cancer (GC) [4,20]. Fewings et al. found that *PALB2* loss of Function (LOF) variants are >7.5-fold more common in HDGC families than in the general population [17]. Its risk can be affected by mutation hotspots, modifier genes, and other unaccounted environmental factors [17,21]. There is limited evidence for a modifier role of *Helicobacter pylori* in *PALB2* mutation carriers.

There are also limited data that patients with *PALB2*-associated non-HDGC may potentially benefit from PARP inhibitor therapies [22]. Therefore, the emergence of *PALB2* as a new familial GC gene may offer double significance for both prevention and targeted and personalized treatment of GC.

Although whole-genome sequencing might identify some additional candidates in regulatory elements or structural variants for many families with HDGC without pathogenic *CDH1* or *CTNNA1* variants, the underlying cause often remains unexplained even after whole-exome sequencing [17]. Future studies of genes associated with disease predisposition could focus on cancer genes with mild to moderate impact, such as *PALB2*, to expand options for personalized therapy.

### 2.2. Spectrum of CDH1 Germline Variants

The gene *CDH1* is located on chromosome 16q22.1 and consists of 16 exons [23]. Germline variants are scattered across the entire gene [24,25]. In the HDGC clinical setting, *CDH1* germline abnormalities are distributed evenly throughout the gene with no apparent genotype–phenotype correlation [26]. While the first germline mutation of *CDH1* was reported in families with HDGC in 1998 [1], more than 100 *CDH1* other germline mutations are described in HDGC at present. The known variants are mainly truncating mutations, usually caused by frameshift, single nucleotide variants, or exon/intron splice site mutations [4]. The missense fraction accounts for 16% of the mutations described so far [24]. The clinical and functional impact of missense mutations still raises controversy among specialists because a full-length protein is preserved in most cases, and regular levels of E-cadherin are usually expressed [24,27].

*CDH1* mutations are inherited in only one allele presenting an autosomal-dominant familial disorder. For initiation of the neoplastic process, downregulation of the second copy of the CDH1 gene or somatic inactivation has to be induced. Epigenetic alterations are suggested to play an essential role in this process. The most common established mechanism leading to biallelic CDH1 inactivation is promoter hypermethylation, while mutation or deletion of the second CDH1 allele is less frequently described [28,29,30,31]. Approximately 27% (41 of 155) of total reported *CDH1* pathogenic mutations have been reported in multiple families, suggesting that germline mutations can either arise from a common ancestor or be the result of novel events at mutational hotspots [4,9,32].

Hansford et al. describe the cumulative risks of GC and breast cancer (BC) for *CDH1* mutation carriers. The cumulative incidence of GC by 80 years was 70% for male participants and 56% for female participants. The risk of BC for female participants was 42% [4]. However, *CDH1* germline variants are associated with multiple disorders beyond elevated susceptibility for diffuse type gastric cancer and LBC. For example, it currently remains undetermined whether colorectal cancer is also a potential manifestation of HDGC [33,34].

### 2.3. E-Cadherin Structure, Molecular Function, and Signal Pathways in Cancer

*CDH1* encodes the 120-kDa transmembrane glycoprotein E-cadherin. E-cadherin is a classical Type I-cadherin that is detected in the adherens junctions of all epithelial tissues [35]. Its structure is highly conserved and consists of an extracellular domain, a transmembrane domain, and an intracellular domain (see Figure 1).

The extracellular domain has five extracellular cadherin repeats (EC1–EC5) with about 100 amino acids each. There are four negatively poled calcium ion binding sides between the five EC-domains each binding three Calcium ions. The EC1s of neighbouring cells interact calcium-dependently in a homophilic manner and mediate rigid cell–cell adhesion [36,37]. Cis interactions of the EC1 domain with the EC2 domain lead to further clustering [38].

The single transmembrane domain bridges the extracellular domain to the intracellular domain. The intracellular domain of E-cadherin has a highly phosphorylated region that binds p120 catenin as well as α catenin via β catenin. α catenin links the actin cytoskeleton to the plasma membrane which is vital for the stability of the epithelial architecture and cellular mechanical signaling.

Truncating and missense variants occur in every domain, truncating domains most frequently in EC2, EC3, EC5 and IC [34]. Missense mutations of the extracellular region of E-cadherin are more likely to result in increased motility of the affected cell than missense mutations of the intracellular region [24].

As shown in Figure 1, via the cadherin–catenin complex, E-cadherin controls cell polarity, cellular stability, cellular homeostasis, cell growth and differentiation. α-E-catenin, the protein of *CTNNA1*, interacts with E-cadherin via β-catenin. Lobo et al. describe mutations in α-E-catenin, which occur in areas of the gene that code for the amino acids 97–148 [39]. These amino acids are forming the N-terminal region of α-E-catenin which binds β-catenin. β-catenin finally binds to the C-terminal region of E-cadherin.

E-cadherin interacts directly or indirectly with many other membrane components (e.g., tight junctions, gap junctions, desmosomes, integrin-based adhesions, and pathogenetic proteins) [36]. The extracellular fragment of E-cadherin can be cleaved by membrane-bound metalloproteinases from the membrane after proteolysis of E-cadherin [40]. Soluble E-cadherin controls multiple signaling pathways via paracrine and autocrine signaling. Its function is modulated by various direct and indirect interactions (e.g., mechanical forces at the junctions, direct phosphorylation, binding and phosphorylation of its catenins, endocytosis, recycling and degradation, multiple signaling pathways, developmental processes) [41].

E-cadherin is a crucial part of signaling pathways like cyclin kinase inhibitor p27-mediated signaling, activation of mitogen-activated kinase (MAPK), rat sarcoma viral oncogene (Ras) and ras-related C3 botulinum toxin substrate (Rac1) signaling, phosphatidylinositol-3-kinase (PI3K)/AKT signaling, HIPPO signaling and epithelial mesenchymal transition [41,42,43].

Loss of E-cadherin weakens the adherens junctions in epithelial tissues and intercellular stability by wnt, the HIPPO pathway, growth factor receptor tyrosine kinases (RTKs) and GTPases [43]. This is the first crucial step towards metastasis, angiogenesis, adhesion and invasion. Uncontrolled activation of intracellular signaling pathways results in uncontrolled proliferation [44].
ijms-23-07821-t001_Table 1Table 1Molecular profiling in HDGC.GenesCorresponding ProteinsCancers in Which the Related Genes ExpressEncoded FunctionsReference*CDH1*E-cadherin Gastric cancer (diffuse type including HDGC), lobular breast cancer, colorectal cancer, hepatocellular carcinoma, squamous cell carcinomas of the skin, neck, and head, esophageal carcinoma, pancreatic ductal adenocarcinoma*Tumor suppressor* and adhesion, adhesion in cell-cell contact [4,10,11,13,14,15,18,24,25,26,27,28,29,30,31,32,33,34,35,38,39,40,41,42,43,44,45,46,47,48]*CTNNA1*Alpha E-cadherinHDGC (HDGC without *CDH1*-mutation, as well), diffuse type GC and colorectal cancer*Tumor suppressor* and adhesion, adhesion in cell-cell contact[16,19,20]*MAP3K6*A serine/threonine protein kinaseHDGC,diffuse and intestinal type GC*Tumor suppressor*[24,25]*BRCA1*, *PALB2*, *RAD51C*Corresponding Proteins on their ownHDGC, breast cancer, pancreatic cancer, pancreatic ductal adenocarcinoma*Regulate* homologous *DNA recombination*[16,17,18]*DOT1L*Histone MethyltransferaseDGCEffect on *DNA Repair*[17,24] *MSH2*Codes for a DNA mismatch repair (MMR) proteinHereditary nonpolyposis colorectal *cancer* (HNPCC) HDGC diffuse and intestinal type gastric cancerComponent of the post-replicative *DNA mismatch repair system (MMR)*[16,17,18]*MET*A protein with an extracellular, transmembrane and a tyrosine kinase domainGastric cancer (intestinal and diffuse type, including HDGC, breast-, prostate-, ovarian- cancer, hereditary papillary renal carcinoma Functions in cellular survival, embryogenesis, cellular migration, and invasion [26]*CD44*A cell-surface glycoproteinHyperplastic polyps, intestinal metaplasia, gastric cancer (intestinal and diffuse type, including HDGC) Cell surface glycoprotein[27]*INSR*Receptor tyrosine kinaseHDGCEffect on tumor cell invasion[17]*FBXO24*F-box proteinDGCTumor driver[17]


### 2.4. Other Contributing Risk Factors for HDGC

Despite a considerable decline in mortality over the last century, GC still remains the fifth cause of cancer-related death worldwide, with an incidence of over 1.08 million new cases [2]. To study its etiology and risk factors remains an important issue as it may allow to identify major targets for primary prevention.

A consortium of epidemiological investigations initiated in 2012 the “Stomach cancer Pooling (StoP) Project” which included over twenty studies in order to analyze the role of lifestyle and genetic determinants in the etiology of GC in general [45]. Patients with gastric cancer from different parts of the world were included (40% from Asia, 43% from Europe, 17% from North America), regardless of the histological subtype; intestinal type as well as diffuse type GC. Among the identified risk factors, *Helicobacter pylori* is still the most prominent, with a significantly higher prevalence in men than women (in men: OR 1.33, 95% confidence interval 1.04–1.70) [45]. Other identified risk factors were cigarette smoking (40% increased risk in smokers versus non-smokers), heavy alcohol consumption (50% increased risk for heavy drinkers compared to never drinkers), meat consumption (especially red, processed, and total meat, with the highest OR observed for an intake of 150 g/day of red meat) and low socioeconomic status [45,46,47].

### 2.5. Histological Alterations in CDH1 Gene Mutation

HDGC has no predilection to occur in a specific area of the stomach. Therefore, total gastric mucosa embedding and mapping should be performed to detect HDGC in prophylactic gastrectomies. This is time consuming and might contribute to a lack of expertise to assess HDGC appropriately. Furthermore, background alterations in *CDH1*-carriers make it even more difficult to discover cancer lesions: Chronic gastritis, foveolar hyperplasia, cystic gland dilatation and epithelial tufting, as well as globoid change, have been described in *CDH1*-carriers [48]. For its histological detection, standardized protocols, WHO and Laurén classification should be used.

Two precursor lesions for HDGC have been described in *CDH1*-carriers—first, “in situ signet-ring cell carcinoma” (SRCC), which is characterized by signet-ring cells (SRC) with hyperchromatic and depolarized nuclei located in glandular basal membranes; second, rows of SRCs with pagetoid growth patterns inside the glandular basal membrane and below the non-neoplastic glandular epithelium and foveola.

In advanced stages of HDGC, SRC and pleomorphic poorly differentiated non-SRCs infiltrate the gastric wall, diffusely in a scattered manner. They are poorly cohesive. The gastric wall appears thickened and rigid. The cells increase in size from the neck gland zone to the surface [8]. Glandular and tubular structures, mucinous areas and rosettes are especially found in areas of lymphovascular invasion and lymph node metastasis [49]. A high KI-67 proliferation index and overexpression of *p53* indicate a highly aggressive phenotype of HDGC cancers [49].

## 3. Implications of *CDH1* Gene Alterations in the Diagnosis and Management of HDGC

### 3.1. IGCLC Diagnostic Criteria for HDGC Syndrome

Due to the relatively low incidence of HDGC, randomized clinical trial data specific to HDGC are lacking. Consequently, recommendations concerning genetic testing criteria rely on consensus expert opinion and evidence as well as observational studies. Over the years, multiple criteria have been developed to facilitate the screening of *CDH1* mutation carriers. The updated guidelines for HDGC of the IGCLC provide recommendations for medical care givers and patients at a risk for the HDGC syndrome. To restore the balance between the accessibility, costs and acceptance of genetic testing and greater identification of pathogenic variant carriers, the HDGC genetic testing criteria have been broadened, mainly through less restrictive age limits [8]. The recommendations of *CDH1* testing according to the IGCLC guidelines are summarized in Table 2.

Furthermore, the IGCLC guidelines intend a better understanding of the nature of the disease and associated risk factors, facilitating decision-making and providing guidance for lifetime management of HDGC families [8].

Due to the association of *CTNNA1* mutations and HDGC [4,15,16,18], IGCLC guidelines recommend that individuals who fulfil criteria for genetic testing but are found to be negative for a *CDH1* variant should be subsequently tested for *CTNNA1* mutation [4,8]. In a study by Clark et al., *CTNNA1* loss-of-function variants were detected on multiplex genetic panel testing (MGPT) with a majority of the individuals diagnosed with gastric or breast cancer [19]. In another study, in *CDH1* mutation-negative index cases, candidate mutations within genes of high and moderate penetrance including *CTNNA1* and *BRCA2* were identified [4].

Although there is only limited evidence that *PALB2* pathogenic gene variants might be associated with an increased risk for GC in *CDH1*-negative patients [50], IGCLC recommends considering *PALB2* testing in the case of unexplained gastric cancer in some families, alongside other genes associated with an increased risk of gastric cancer (eg. *ATM*, *BRCA2*, Lynch syndrome genes, *APC* and *TP53*) [8].

### 3.2. Genetic Counseling and Genetic Testing for HDGC Syndrome

The clinical phenotype of HDGC shows considerable heterogeneity concerning the type of cancer and age of onset [4,51]. In some families, GC onset has been described in a very young patient’s age [10]. In other families, the main cancer phenotype is LBC and either no other family members or only older family members are affected by HDGC. Some reports also suggest an association with colorectal cancer [34].

The age of testing at-risk relatives of index patients should take into consideration the earliest age of cancer onset in that specific family. Testing from the late teens or early 20ies is favored in families with early-onset DGC. Because of the autosomal-dominant inheritance and high penetrance of *CDH1* germline mutations [52,53], genetic testing guided by personal history, family history, detailed three-generation pedigree analysis, and other risk models are currently used to identify the individuals at risk for HDGC syndrome [8].

Retrospective data provide evidence that early genetic counseling and detection of *CDH1* mutation in asymptomatic carriers improves survival in HDGC [54,55]. However, a study by Xicola et al., conducted with 113 *CDH1* pathogenic variant probands and 476 relatives, demonstrated that, in unselected *CDH1* pathogenic variant carrier families who do not fulfil the criteria of HDGC, the GC risk is low [52]. In addition, the age at first diagnosis was higher than previously reported in families pre-selected for HDGC criteria. A substantial proportion of families with cancer limited to breast do not present with any GC [52].

Genetic testing includes two main clinical approaches: single-gene testing or multiplex genetic panel testing (MGPT). Both tests can be performed on DNA extracted from blood or buccal samples except for patients who have received allogenic bone marrow transplantation or a recent diagnosis of hematologic malignancy. In these patients, DNA from a fibroblast culture is the preferable tissue sample.

The spread of MGPT has led to the identification of *CDH1* and *CTNNA1* germline pathogenic and likely pathogenic variants in individuals without a personal and family history suggestive of HDGC. This issue poses challenges regarding the clinical management [19,56].

The *CDH1* gene analysis includes the examination of the entire coding sequence by PCR amplification and DNA sequencing, including intron-exon boundaries [57,58,59]. The sequence analysis by PCR based Sanger sequencing and direct sequencing allows for detecting small intragenic deletions/insertions and missense, nonsense, as well as splice variants of *CDH1* and other germline mutations. Exonic or whole-gene deletions and duplications are detected by quantitative PCR, long-range PCR, multiplex ligation-dependent probe amplification (MPLA) and chromosomal microarray analysis (CMA) [60].

A study reported by Molinaro et al. supports a multimethod approach for *CDH1* testing using a series of molecular methods, including DNA sequencing, MPLA, single-nucleotide primer extension, bisulfite sequencing, reverse-transcription PCR and bioinformatics tools [61]. It demonstrates that both DNA and RNA analysis are required to increase the detection rate of pathogenic mutation. The number of patients without a clear molecular diagnosis could be reduced by this approach [61]. Another study suggests that a multigene analysis is important to detect germline mutations and genetic variants in patients with cancer at multiple sites in the family [62].

Bio-imaging analysis of in situ fluorescence microscopy has been described to quantify missense *CDH1* variants [63]. The study illustrated that this method could be applied to evaluate the pathogenicity of the *CDH1* mutant variant as a complementary approach and to detect a wide range of proteins and diseases characterized by aberrant protein expression or trafficking deregulation.

The updated variant curation guidelines published in 2018 by the American College of Medical Genetics and Genomics and the Association for Molecular Pathology (ACMG/AMP) provide a framework to assess variant pathogenicity of the germline *CDH1* sequence variants. They facilitate a better detection and classification in *CDH1* variants and thus identification of individuals and families that will benefit from cancer surveillance and risk-reducing surgery [64].

With the help of the rapid development of genetic testing technologies and data analysis tools over the last few years, genetic testing has become more efficient, enabling the discovery of susceptibility genes for HDGC.

However, in the assessment of the appropriate clinical management and decision-making, a multidisciplinary team consisting of gastroenterologists, genetic counselors, medical geneticists, pathologists and oncological psychologists is of key importance [65]. Due to the variable penetrance of the *CDH1* or *CTNNA1* mutation, unaffected carriers from HDGC families face difficult decisions and can be assisted best through education and interactions with counseling by an informed interdisciplinary team [66].

Genetic counseling should include the initial evaluation of personal history, family history and a three-generation family pedigree as well as any history of cleft lip or cleft palate and histopathological confirmation of cancer diagnosis or precursor lesions [8].

Due to the high psychosocial impact on quality of life and long-time sequelae, a discussion on lifetime risks of HDGC or LBC, as well as the possibility of undergoing PTG or other options of surveillance such as endoscopy, should be provided [55,67,68]. The counselee should also be informed about the potential significance of the test results for the family, and about reproductive options, such as the availability of genetic testing through prenatal and preimplantation genetic diagnosis [12].

### 3.3. Management of CDH1 Mutation Gene Carriers

The heterogeneity of the clinical phenotype in HDGC makes a balanced screening strategy necessary. For risk stratification, it may help to identify different *CDH1* mutation types. Lo et al. describe associations of *CDH1* germline variant location and cancer phenotype in families with HDGC [34]. The frequency of truncating germline *CDH1* variants varied across functional domains of the E-cadherin receptor gene and was highest in linker and PRE regions. Families with truncating *CDH1* germline variants located in the PRE-PRO region were six times more likely to have family members affected by colorectal cancer (OR 6.2) compared with germline variants in other regions. Variants in the intracellular E-cadherin region were protective for cancer at a young age (OR 0.2) and in the linker regions for breast cancer (OR 0.35). Different *CDH1* genotypes were associated with different intracellular signaling activation levels including different p-ERK, p-mTOR, and β-catenin levels in early submucosal T1a lesions of HDGC families with different *CDH1* variants [34].

These data suggest that the location and type of *CDH1* germline variants may help to identify families at increased risk for concomitant cancers that might benefit from individualized surveillance and intervention strategies. However, there is no strong evidence that the risk of other cancers is significantly increased in individuals with a particular *CDH1* pathogenic variant [4,61,69].

For female *CDH1* mutation gene carriers, breast surveillance with annual MRI should start at the age of 30 years [70]. The benefit of adding mammography to MRI in young women remains uncertain [71]. A woman with a *CDH1* pathogenic variant may choose breast-conserving surgery depending on the individual risk profile [72].

Current data suggest that a germline mutation in the *CDH1* gene is present in a proportion of signet-ring carcinomas in the colorectum. A subset of these tumors could be familial. However, the loss of E-cadherin staining in the absence of *CDH1* mutations also suggests a role for environmental factors in a subset of these tumors [73,74,75]. Interestingly, the colorectal adenocarcinoma associated with HDGC syndrome is not necessarily signet ring cell carcinoma [76]. However, there is currently no evidence to suggest that the risk of colorectal cancer in *CDH1* mutation carriers is significantly elevated and up to now there are insufficient data to give recommendations on colorectal cancer screening [69].

In *CDH1* and *CTNNA1* gene mutation carriers, the question arises as to whether endoscopic surveillance for HDGC might be sufficient or surgery be indicated. A significant challenge in HDGC syndrome is balancing the high risk of GC against the morbidity of total gastrectomy that includes dumping syndrome, weight loss, malnutrition and depression.

Castro et al. describe that, in their cohort of patients with *CDH1* mutation gene carriers, gastroscopy with targeted and random biopsies had low sensitivity. In that cohort, 332 random and 22 targeted biopsies were needed to identify a single SRCC focus. Total gastrectomy was performed in 13 patients and SRCC foci were identified in 12 surgical specimens [77]. In contrast, Dieren et al. describe a higher sensitivity of gastroscopy regarding SRCC in *CDH1* mutation gene carriers. Out of 30 patients who underwent gastrectomy, at least one SRCC lesion was found in 26 of 30 gastrectomy specimens. In 18 of these 26 surgical specimens (69%), SRCC had been identified by endoscopic biopsies. The low number of SRCC detected through random sampling stresses the importance of targeted biopsies [78].

Another study by Mi et al. reports on endoscopic surveillance for HDGC according to *CDH1* mutation status. Eighty-five individuals fulfilling HDGC criteria were included and underwent 201 endoscopies. Fifty-four patients (63.5%) tested positive for *CDH1* mutation. SRCC yield was 61.1% in *CDH1* mutation carriers compared with 9.7% in non-carriers, and mutation-positive patients had a 10-fold risk of SRCC on endoscopy compared with those with no mutation detected (*p* < 0.0005) [79].

The reported studies show that a combination of random and targeted therapies increases the sensitivity of gastroscopy in *CDH1* mutation gene carriers and that endoscopic surveillance is an option especially with high resolution and high-definition endoscopes with advanced endoscopic imaging [80]. As *CDH1* mutation carriers harbor an approximately 70% lifetime risk of gastric cancer in men and 56% in women [4], the critical question remains as to whether endoscopic surveillance only unnecessarily delays a gastrectomy. An ongoing study (NCT04253106) examines the role of liquid biopsies in blood and gastric fluid in asymptomatic carriers who decline gastrectomy. This might be a complementary approach to endoscopic surveillance for these patients in the near future.

These studies further stress that one of the major problems is the variable penetrance of *CDH1* gene mutation carriers and other distinct gene mutations as well to a clinical phenotype and HDGC. This issue might be addressed in a pooled data analysis of the data that are already available including all patients that underwent PTG. Furthermore, this aspect might be addressed in prospective, multicentric register studies. As we are dealing with a rare disease but tremendous impact on patients and families, all patients with manifestation of HDGC, LBC but also *CDH1* gene mutation carriers without penetrance of the malignant phenotype should be prospectively included in such registers.

### 3.4. Therapy Strategies for HDGC

Endoscopic surveillance and/or gastrectomy are the pillars of management for patients with HDGC. In case of locally restricted expansion, surgical resection is indicated. In stage IB-III, resection is combined with perioperative or adjuvant chemotherapy in accordance with sporadic, non-HDGC [81]. Local progressed, non-resectable or metastatic disease should be treated with chemotherapy. The FDA approved Pembrolizumab combined with Trastuzumab and Chemotherapy as first-line treatment in locally advanced unresectable or metastatic HER2-positive gastric or gastroesophageal junction adenocarcinoma and for recurrent locally advanced or metastatic gastric or gastroesophageal junction cancer with PD-L1-expression. For patients without HER2-overexpression but with high PD-L1 expression on the tumor cells and infiltrating immune cells (PD-L1 CPS > 5%), Nivolumab in combination with chemotherapy is recommended and approved by FDA and EMA.

Unfortunately, there is currently no specific molecular genetic therapy available for HDGC. Due to the rarity of the tumor disease, there are currently no studies investigating specific molecular treatment strategies specific for HDGC. Drug screening studies in isogenic *CDH1*−/−-mutant mammary epithelial MCF10A and c.1380delA *CDH1*-mutant gastric cancer cells raise the hope that HDGC might respond well to PI3K, mTOR, or ALK/ROS-1 like tyrosine kinase inhibition [25]. These inhibitors are already widely used in breast cancer and non-small cell lung cancer (NSCLC), among others. New trials on gastric cancer focus on microRNAs, suppressor-tRNA, antibodies targeting tyrosine kinase receptors, immune checkpoints (PDL-1, CTLA-4, TIM-3, LAG-3), and cytokines (TGF-β, Il-6), cell therapy, and virotherapy [82].

## 4. Conclusions

The clinical phenotype of HDGC shows considerable heterogeneity with regard to the type of cancer and age of onset, rendering the identification of patients with a high pretest probability of HDGC challenging. Due to the rapid development of genetic testing technologies and data analysis tools over the last few years, genetic testing has become more efficient, enabling the discovery of susceptibility genes for HDGC. Screening for *CDH1* and *CTNNA1* mutation (alongside other genes associated with increased risk of gastric cancer in the case of unexplained gastric cancer in some families) should be performed in individuals who fulfil criteria for genetic testing, followed by a thorough evaluation and interdisciplinary approach to establish the best therapeutic approach (endoscopic surveillance versus prophylactic gastrectomy). In the assessment of the appropriate management and decision-making, a multidisciplinary team consisting of gastroenterologists, surgeons, genetic counselors, medical geneticists, pathologists, and oncological psychologists is therefore of key importance.

## Figures and Tables

**Figure 1 ijms-23-07821-f001:**
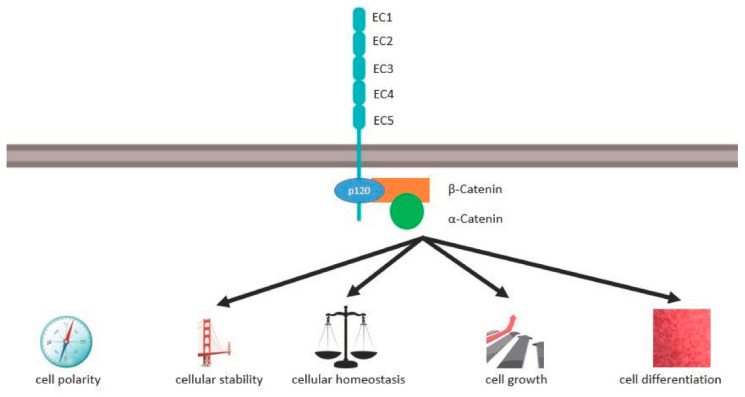
Role of E-cadherin for cellular homeostasis, polarity and differentiation.

**Table 2 ijms-23-07821-t002:** IGCLC criteria for CDH1 testing (according to Blair et al., hereditary diffuse gastric cancer: updated clinical practice guidelines, 2020).

Family Criteria (1st or 2nd Degree Blood Relatives of Each Other) ^1^	Individual Criteria
≥2 cases of gastric cancer in family regardless of age, with at least one DGC ≥1 case of DGC at any age and ≥1 case of LBC < 70 years in different family members ≥2 cases of LBC in family members < 50 years	DGC < 50 years DGC at any age in individuals of Māori ethnicity DGC at any age in individuals with a personal or family history (1st degree) of cleft lip/cleft palate
	History of DGC and LBC, both diagnosed < 70 years Bilateral LBC, diagnosed < 70 years Gastric in situ signet ring cells and/or pagetoid spread of signet ring cells in individuals < 50 years

^1^ If there are no living affected relatives, consider tissue testing (tumor or normal) from an affected deceased relative. If these options are not possible, consider indirect testing in unaffected family members.

## Data Availability

Not applicable.

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
