# Peer review of "Hereditary Diffuse Gastric Cancer: Molecular Genetics, Biological Mechanisms and Current Therapeutic Approaches"

_ijms, 2022, doi:10.3390/ijms23147821_

Round 1

Reviewer 1 Report

The authors presented a literature review covering genetic variants and therapeutic approaches in hereditary diffuse gastric cancer.  The review is thorough with regard to literature citations. There have been previous reviews covering HDGC and it is not always clear in the text what findings since the last previous review have contributed to the diagnosis and treatment of HDGC.  Also, in some cases, factors affecting GC in general are not described as to how they apply specifically to HDGC.  Overall, the review is informative on this topic.  I have only a few minor comments:

pg. 2, line 55, “mutation is diagnosed”:  Mutations (variants if germline) are detected, not diagnosed.

pg. 2, line 70:  The preferred term is driver mutations, rather than driving mutations.

 pg. 2, line 71:  Do you mean mutations only in cancer-predisposing genes or driver mutations in other genes as well? 

pg. 4, Section 2.3: Is there any association of missense alterations within EC1 or intracellular domain?  Do these missense variants correlate with the presence of disease?  How frequently do the truncating variants eliminate these domains? Do alterations in catenin alpha-1 occur in areas of the gene that code for amino acids that associate with CDH1 interaction domains?

pg. 4, lines 128-129: It has been reported that CDH1 promoter methylation is responsible for less than 1% of CDH1 inactivation, while deletion is responsible for 3.8%.  Does the “most established” mechanism refer only to biallelic inactivation?

pg. 4, line 160: What protease performs cleavage of E-cadherin?

pg. 5, line 175: Does the “diffuse” description of HDGC relate to metastasis?   

pg. 5, line 181ff: Do these risk factors for GC affect penetrance of inherited HDGC?

pg. 5, line 194: Please define “background alterations”.  Does this refer to precursor lesions or the tissue abnormalities mentioned in lines 195-196? Figures showing examples of these lesions would be helpful.

pg. 7, line 280: A diagram of the gene and protein domains with locations of variants would be helpful with regard to description of PCR testing methodologies.  It would also aid in relating the pathogenicity of the variants.

pg. 8, line 336:  Cumulative risk curves may be more demonstrative than the text descriptions.

pg. 8, line 340:  Previous reviews of HDGC include ethnicity of carriers.  It might be worth addressing the relationship (or lack thereof) of ethnicity for carriers or HDGC incidence.

Author Response

Please see the attached file as point-to-point response.

Reviewer 2 Report

Lidia-Sabina Cosma and collaborators performed an accurate and elegant review manuscript of hereditary diffuse gastric cancer. 

I believe this review is well done, I have only few minor comments to be considered by the authors:

1. Gramathical: genes should be written in italics. 

2. Table 1 summarizes the molecular profiling in HDGC, although in column 3 is listed the cancers related with specific genes. In this sense, there are genes related to gastric cancer,  not only HDGC, but also intestinal subtype, and this should be specify.

3. Point 2.4 title "Other contributing risk factors for HDGC". This paragraph describe the main risk factors associated with GC, including also the results of StoP project. This risk factors are more related to intestinal GC, and I suggest to the authors to clarify if this results are also associated to the HDGC, as it says in the subtitle. 

4. It would be very interesting if authors delve into future perspectives.

5. Finally I suggest to add a figure, may be with the role of CDH1. 

Author Response

please see the attached file as point-to-point-response.
